# Oxidative Stress and NADPH Oxidase: Connecting Electromagnetic Fields, Cation Channels and Biological Effects

**DOI:** 10.3390/ijms221810041

**Published:** 2021-09-17

**Authors:** Christos D. Georgiou, Lukas H. Margaritis

**Affiliations:** 1Department of Biology, Section of Genetics, Cell & Developmental Biology, University of Patras, 10679 Patras, Greece; c.georgiou@upatras.gr; 2Section of Cell Biology and Biophysics, Department of Biology, National and Kapodistrian University of Athens, 26504 Athens, Greece

**Keywords:** EMF, voltage-gated calcium channels, cation channels, ROS, DNA damage, carcinogenesis

## Abstract

Electromagnetic fields (EMFs) disrupt the electrochemical balance of biological membranes, thereby causing abnormal cation movement and deterioration of the function of membrane voltage-gated ion channels. These can trigger an increase of oxidative stress (OS) and the impairment of all cellular functions, including DNA damage and subsequent carcinogenesis. In this review we focus on the main mechanisms of OS generation by EMF-sensitized NADPH oxidase (NOX), the involved OS biochemistry, and the associated key biological effects.

## 1. Introduction

Oxidative stress (OS) generation, expressed by the production of reactive oxygen/nitrogen species (ROS/RNS), has been linked to the exposure of animals and cells to man-made EMF. The related key experimental studies of the last decade have been presented elsewhere [1]. However, the biochemical specifics of the biological mechanisms involving the generation of OS are unclear, with the most well studied mechanisms being those involving ion channels. The main hypothesis underlining these mechanisms portrays EMF (RF 900, 1800, 1900, 2450 MHz, ELF 0–3000 Hz) exerting their oscillatory forces on every free ion on both sides of any biological membrane, thereby causing the ions, cations, in particular, to pass through at abnormal rates. Such abnormal cation movement can alter the biochemical properties of membranes and can cause deterioration of cation channel functions, especially those of the voltage-gated (VGC) ones [2,3,4,5,6,7,8]. These, in turn, can trigger an increase of OS, leading to the impairment of most cellular functions and DNA damage [9,10] as well as to numerous associated diseases including carcinogenesis [11,12,13]. In this review, we examine (i) the main mechanisms of OS generation by EMF-induced VGCs and their association with the ROS-generating NADPH oxidase (NOX) and the voltage-gated calcium channels (VGCC), (ii) the involved OS biochemistry, and (iii) the VGC-OS-originating key biological effects. We conclude with the need for research to unequivocally prove that non-ionizing EMF can directly generate ROS.

## 2. EMF Effect on Membrane Function and Cation Channels Can Induce OS

Disruption of the electrochemical balance and cation channel function of the cell membrane by EMFs and the generation of OS can involve the following mechanisms:

EMF disturbance of membrane function and cation channels: Abnormal function of cell membranes and ion channels by EMFs primarily take place by means of the following mechanisms:EMFs can act on membrane-associated free ions by means of a forced-vibration mechanism, which acts on the voltage-gated channels using the vibrations that EMFs cause on the transported ions [2,3]. These are the Κ^+^ leak channels, the Νa^+^ leak channels, and the leak channels of other cations such as Ca^2+^. Briefly, the open or closed state of these channels is determined by the electrostatic interaction between the voltage sensors of the channels and the transmembrane voltage, and they change between their open and closed state when the electrostatic force on the electric charges of their voltage sensors exceeds a critical value. Alternating electric fields exert a periodic force on all of the free ions passing across voltage-gated channels and can distort the electrochemical equilibrium state of every oscillating ion due to its site-displacement from the developed electrochemical gradient. This results in a periodical displacement of the electric charge, which will exert force on every fixed charge, such as those existing on the voltage sensors of the voltage-gated channels. Specifically, voltage-gated channels contain transmembrane *a*-helices in parallel, forming pores through which ions pass in their dehydrated state. The voltage sensors of these channels are four positively charged symmetrically arranged transmembrane *alpha*-helical domains (designated S4). It is known that small voltage changes of about 30 mV in the membrane potential are able to gate this kind of channel [14,15]. Such a change can be caused by the displacement of a single ion by 10^−12^ m from the electric field of the EMF and in the vicinity S4 of the voltage-gated channels. Hence, EMF-induced oscillating ions can disturb the electrochemical balance of the membrane via the gating of such channels, and those ions crossing such channels can change their normal positions and can produce a false signal for the gating such channels with their charge. This mechanism can also explain the biological action of oscillating magnetic fields by replacing the force of the electric field with the force exerted by an alternating magnetic field and also by accounting for the induced electric field, which is always generated by the pulsed magnetic one. The mechanism concludes that oscillating electric or magnetic fields with frequencies lower than 1.6 × 10^4^ Hz (ELF and VLF fields) can be bioactive, even at very low intensities [2,16]. It is also claimed that pulsed EMFs can even further amplify their biological action compared to continuous EMFs [16,17,18].In the other mechanism [7,8], low intensity EMFs cause abnormal hypersensitivity on voltage-gated calcium channels (VGCC), with consequent increases in intracellular Ca^2+^ (and OS) acting on the structure of the aforementioned voltage sensor (and its 20 electrically charged groups that are equally distributed on its 4 *alpha*-helixes), which actually regulates the opening of the channel. They transverse the lipid bilayer of the membrane and their disturbance by the EMFs relies on the following very important and distinct reasons: The EMF forces on the voltage sensor are 20 times higher since they act on all 20 charges, and according to Coulomb’s Law of physics, they are inversely proportional to the dielectric constant of the fatty part of the membrane, which is about 120 times less than the forces on the charges in the aqueous parts of the cell. Thus, the forces exerted on the 20 electrically charged groups of the VGCC’s voltage sensor by the electric field of the RF EMF will also be about 120 times stronger (or 2400 times stronger, accounting for all 20 charges), which is only because of the dielectric constant of the fatty region of the cell membrane where the voltage sensor’s 20 charges are located. A third, even more important, reason is based on the fact that the cell membrane has a very high electrical resistance, which acts as amplifier of the electrical gradient (the difference in electrical charge across the cell membrane), amplifying it by about 3000 times. Combining these three distinct reasons, it is implied that the total amplification of the exerted forces by the RF EMF electric fields on the VGCC voltage sensor’s 20 electrical charges is equal to 20 × 120 times (due to the *dielectric constant* of the fatty inner space of the membrane) × 3000 times (due to the *electrical gradient* of the membrane), totaling 7,200,000 times. That is, the forces exerted on the VGCC voltage sensor by the RF EMFs are about 7.2 million times stronger than those in the electrically charged groups that are in the hydrophilic environment of our cells, which is where the safety guidelines for the RF EMF are set by ICNIRP.

EMFs act via the activation VGCC in the plasma membrane, producing excessive Ca^2+^, which leads to the pathophysiological effects associated with ROS, such as nitric oxide (NO), superoxide radical (O_2_^•−^), and peroxynitrite (ONOOH) [6]. Studies on the mechanisms related to VGCC and to the associated pleiotropic effects are presented elsewhere [1].

OS generation by EMF disturbance of membrane function: NADPH oxidase (NOX) is a plasma membrane enzyme and has activity that is very likely associated with the increased generation of OS, upon membrane voltage-gated channel function interaction with EMF. NOX (found in the plasma membrane the neutrophil white blood cells) normally generates OS (or respiratory burst) to eliminate invading microorganisms by means of the production of reactive oxygen species (ROS) [19]. Specifically, NOX produces intracellular O_2_^•−^, which is dismutated to H_2_O_2_, and can be used by myeloperoxidase as substrate for the two-electron oxidation of Cl^−^ to the cytotoxic and highly reactive ROS oxidant hypochlorous acid (HOCl) (Figure 1) [20]. Alternatively, H_2_O_2_ can move extracellularly or to any intracellular site, where it can be converted to the most potent biological oxidant hydroxyl radical (^•^OH) (Table 1, reaction 1), which can oxidatively damage all biological molecules, including DNA (see below).

NOX is the only membrane bound enzyme with channel voltage-gating associated ROS production increase by EMF (3-fold in HeLa plasma membranes exposed to 875 MHz at 0.240 mW/cm^2^ for 10 min; i.e., ~4-fold below the ICNIRP limit). However, this EMF-induced activity increase does not correspond to a proportional increase of NADPH oxidation but is instead displayed in comparison to its inhibition by the oxidase’s specific inhibitor diphenyleneiodonium (DPI) [21]. Nonetheless, this EMF effect on NOX may be indirect and is exerted on an oxidase-associated voltage-gated channel that is blocked by DPI, given that DPI acts not only as an inhibitor of the oxidase but also as a cation channel blocker [22]. This suggests that the NADH oxidase’s direct response to particular EMF exposure may not be conclusive [23] and that its activity is indirectly associated or controlled by neighbouring cation voltage-gated channels. There are studies that support such cation voltage-gated control of NOX activity. It has been known that electron flux within NOX (from NADPH through a redox chain comprising FAD and two haems) is electrogenic and that it rapidly depolarizes the membrane potential by ~100 mV during respiratory burst, necessitating the need of proton (H^+^) channels for the reduction of extracellular O_2_ to O_2_^•−^ by the oxidase (Figure 1) [24,25,26]. Indeed, it has been shown that the activities of NOX and voltage gated H^+^ channels present (especially the H^+^ channel Hv1) in many cells (immune cell types, including neutrophils, eosinophils, B-lymphocytes, T-lymphocytes, macrophages and microglia) appear to be coordinated [27]. The voltage-gated H^+^ channels Hv1 contribute the majority of the charge compensation required for high-level NOX-dependent O_2_^•−^ [28,29]. Concomitant K^+^ release elimination only occurs at maximum O_2_^•−^ production, while the reversed membrane potential inhibits Ca^2+^ entry (e.g., by VGCC), thereby preventing the cells from becoming overloaded with Ca^2+^. Thus, a clear correlation exists between the rate of O_2_^•−^ production by NOX and the extent of plasma membrane depolarization and inhibition of Ca^2+^ entry. Disturbance of this limiting mechanism by EMFs may contribute to OS-associated disease [30].

Besides the voltage-gated H^+^ channels Hv1 coordinating/associating with NOX activity, the oxidase itself may act directly as a proton channel due to its gp91^phox^ transmembrane component (Figure 1), which has been demonstrated to be capable of acting as an oxidase-associated H^+^ channel and functioning as a voltage-gated H^+^ conductance pathway [31].

Nonetheless, electric fields exert their oscillatory force on any subcellular membrane as well, especially if the abnormal function of their ion channels may be associated with increased OS. This is especially true for the mitochondrial ion channels/transporters since they also function as sensors and regulators of cellular redox signaling [32,33]. An abnormal OS increase can take place at two main sites in the oxidative phosphorylation electron-transport chain; at the proton transporter complex I (NADH dehydrogenase) and at complex III (cytochrome *bc_1_*), leaked single electrons reduce O_2_ and generate O_2_^•−^ in the mitochondrial matrix, which, in turn, may activate inner-membrane anion channels and may (through O_2_^•−^ dismutation to cell membrane permeable H_2_O_2_) affect neighboring mitochondria or cytosolic targets as well [32]. The inner mitochondrial membrane ion channels that have been intensely researched are the Ca^2+^ Uniporter (the primary route of Ca^2+^ into the matrix) and the Permeability Transition Pore (PTP; has a quite high conductance, ~1 nS; is a non-selective transporter of solutes up to 1500 Da), which are considered to be among the channels that contribute the most to mitochondria dysfunction under stress. An increased Ca^2+^ flow in the mitochondrial matrix by the Ca^2+^ Uniporter stimulates oxidative phosphorylation (thus, OS increase) in one or more sites of the electron transport chain as well as in sites of the Krebs Cycle. PTP is considered to be activated by depolarization, Ca^2+^, and OS [32]. Irrespective of their cellular sites, EMF-induced cation channels can act as ROS sources for the redox modulation of cation voltage-gated channels in general. Intracellular redox status has been known to significantly alter the gating properties of Ca^2+^ channels (L-, T-, P/Q-type) and Na^+^ and K^+^ channels [34], with the oxidative modulation of voltage-gated K^+^ channels being the most studied [35].

## 3. Biochemistry of ROS

The main cytotoxic ROS is ^•^OH and its production involves the transition metal-catalyzed oxidation of H_2_O_2_ (Fenton reaction [36]), which, once formed, can freely move intra-/extra-cellularly and cross the nuclear and sub-cellular membranes, and can therefore reach DNA. The ROS-associated main source of H_2_O_2_ is the dismutation of O_2_^•^^−^ (which can be generated either by the plasma membrane NOX or by O_2_ after the uptake of single electrons that have been leaked during oxidative phosphorylation). In turn, H_2_O_2_ can be converted to ^•^OH by free Fe^2+^ (or Cu^+^) via the Fenton reaction (Table 1, reaction 1), and its association with the cell’s response to EMF in relation to DNA damage has been also suggested [37]. To sustain the production of ^•^OH, Fe^2+^ (and Cu^+^) must be regenerated via reduction by O_2_^•−^ or H_2_O_2_ (Table 1, reactions 2 or 3, respectively) [38,39,40] and by ascorbic acid (AH_2_) (Table 1, reaction 4) [41].

Nonetheless, ^•^OH is not the sole product of the Fenton reaction in the presence of bicarbonate (HCO_3_^−^, even present at low concentrations, i.e., under physiological conditions), since—in the presence of the Fenton reactants Fe^2+^ and H_2_O_2_—the carbonate radical anion (CO_3_^•−^) is the active oxidizing product [42]. CO_3_^•−^ can then migrate over long distances and can reach DNA, where as a potent one-electron oxidant (as ^•^OH is also), it can generate base radical cations, which ultimately generate 8-oxo-7,8-dihydroguanine (since guanine is the most oxidatively sensitive base in DNA [43]).

However, ^•^OH can be produced even without transition metal mediation. That is, it can be produced either via the oxidation of the ascorbate monoanion (AH^−^) by H_2_O_2_ (Table 1, reaction 5) [44] or by a sequence of reactions involving the reactive nitrogen species (RNS) nitric oxide radical (^•^NO). The latter, upon reaction with O_2_^•−^, produces the biotoxic RNS ONOOH (Table 1, reaction 6), the decomposition of which, besides ^•^OH, produces the RNR radical nitrogen dioxide (^•^NO_2_) (Table 1, reaction 7) [45,46].

Considering O_2_^•−^ and H_2_O_2_ balance in vivo, it has been suggested that cell survival is promoted by an intracellular increase of O_2_^•−^, which is a conjugated base that sustains a cytosolic pH that is mildly alkaline. On the other hand, apoptosis is promoted by intracellular H_2_O_2_ production accompanied by a decrease in O_2_^•−^ levels and concomitant cytosolic acidification, while an excessive production of both O_2_^•−^ and H_2_O_2_ induces necrotic cell death [47].

## 4. EMF-Induced OS, and Biological Effects

The primary biological effect of EMF involves increased OS, which is demonstrated by the elevated levels of ROS/RNS in exposed experimental models. EMFs have been reported to be associated with many downstream effects that can lead to cancer. These involve DNA damage (single/double strand DNA breaks, oxidized DNA bases); increased ornithine decarboxylase, NF-kappa B, tumour promotion (via degradation of gap junction proteins, and DNA breaks/gene amplification), tissue invasion/metastasis (via increased tight junction protein degradation), CaMKII (via protein oxidation), cellular oncogene transcription (by Ca^2+^), and CaMKII (via produced Ca^2+^); lowered melatonin levels; activation of Ca^2+^-dependent phosphatidylserine flippase and c-src (by calcium/calmodulin); and calpain activation of tumour migration, tissue invasion, and metastasis [7].

Through these events, several indirect changes may occur that can alter the physiology, e.g., of brain cells, and can cause translation–transcription interference [48] (through protein conformation changes [49]), cellular metabolism dysfunction, and membrane dyspermeability [50]. The association of the plasma membrane and mitochondrial cation channels with OS-induced damage and their potential participation in pathological liver conditions has been reviewed elsewhere [11].

It has been proposed that RF EMFs invoke a mechanism that involves the rapid activation of ERK/MAPKs (mitogen-activated protein kinase), which is mediated in the plasma membrane (of Rat1 and HeLa cells) by NADH oxidase, followed by a rapid generation of ROS [21]. In this mechanism, ROS accumulation-induced OS leads to a signal transduction pathway involving ERK kinase activation [21,51], cation channel disturbance [21,52], heat shock protein activation [21,48,53], and enzyme conformation change [54]. This, in turn, may affect gene overexpression/suppression [53], possibly through transcription factor activation/deactivation [55], which is random, given the non-targeted impact of EMF. As an end result, stress response related functions may be triggered, including the altered expression of proteins, some of which are related to neural plasticity, whereas others are involved in general metabolic processes [56].

Recent evidence indicates that ROS/RNS-induced OS is among the main intracellular signal transducers, sustaining lysosomal autophagy and nuclear DNA damage response [57,58]. In general, DNA base damage by ROS involves the formation of single lesions in the pyrimidine and purine bases, intra/inter-strand cross-links, purine 5′,8-cyclonucleosides, and DNA-protein adducts formed by the reactions of the 2-deoxyribose moiety and/or the nucleobases with ROS such as singlet oxygen (^1^O_2_), ^•^OH, and HOCl [59]. Even the oxidant role of the ROS singlet oxygen, ^1^O_2,_ on DNA damage was suggested three decades ago [60], and its physiological consequences have been highlighted since ^1^O_2_ is produced in human tissues and human neutrophils [61]. However, ^•^OH is considered as the most potent oxidant of DNA, as verified by extensive studies on the oxidative modifications on DNA bases, strand breakage, and DNA structure [62,63,64]. Additionally, ROS production by mitochondria leads to mtDNA exposure to OS and subsequent damage (being higher compared to that of nucleic DNA [65]), the accumulation of which leads to further increased production of ROS/RNS and damaged mitochondria. The elimination of the latter by lysosomes, so called mitophagy, has been reported to take place both in yeasts and mammals [66]. The mechanisms of oxidative DNA damage leading to mutations and disease are also well known [67,68], and OS-induced DNA damage [69,70] has been reviewed elsewhere [71].

Experimental studies on RF EMF-induced OS effects in various cellular systems, molecules or model organisms have been reviewed by Yakymenko et al. 2016 [72]. Indicative ones are on sperm [73,74] and *Drosophila* [75] (body/ovaries [76]). Near-field EMF GSM (at 900 MHz, ‘‘modulated’’ via speaking vs. nonspeaking emission) decrease the reproductive capacity of *Drosophila* by 50–60% vs. 15–20% (after exposure for 6 min/day during only the first 2–5 days of adult life) [77], the oogenesis of which has been proposed as a biomarker for EMF sources [78]. Extending research in the chronic whole body exposure of mice Balb/c to EMF (GSM at 900 MHz for 3 h/day and by wireless DECT base for 8 h/day), proteome changes (overexpression/downregulation) were recorded in the frontal lobe, hippocampus, and cerebellum [56]. More distinct changes in all of these brain regions were those for synapsin-2 and NADH dehydrogenase, while for some others are indicative of OS increase in the nervous system [79]. In human neuroblastoma cells, low-level GSM EMFs cause alterations on Amyloid Precursor Protein processing and cellular topology, and changes in monomeric *alpha*-synuclein accumulation and multimerization, which can happen concurrently by means of the induction of OS and cell death, which are possibly linked to Alzheimer’s and Parkinson’s diseases [80]. Neurological abnormalities by RF EMF (GSM) are extended to effects on transient and cumulative memory impairments [81] and on short-term memory in mice (by impairing them to pass successfully the Object Recognition Task [82]), possibly due to disturbance of cation channels, particularly that of Ca^2+^ (as also suggested by the EMF effect on the calcium binding protein [83]), and to proteome expression changes in the mouse brain hippocampus and other memory-related brain regions [56].

## 5. Conclusions

On the basis of the above findings, an EMF mechanism can involve ROS formation due to membrane and voltage-gated cation channel function deterioration [2,3,7,8] followed by stress activation and heat-shock protein overexpression [56], which may be associated with behavioural and physiological effects such as blood–brain barrier disruption, memory malfunction, changes in gene expression [53], autophagy, apoptosis [53,84] (especially due to modulation [85]), lifespan reduction, DNA damage, and cancer [18].

## 6. Theory and Research Perspectives for a Conclusive Linking of EMFs with ROS/RNS

EMF induction of OS via increased concentration of free radicals, has been challenged (by ICNIRP) mainly due to (i) the claimed non-ionizing nature of EMF (ELF/RF), where no covalent bonds are broken at non-thermal intensities, or so the argument goes, and because (ii) the measurement of OS is performed by non-specific methods. Indeed, OS is measured, either by methods that are not specific to the identification of generated free radicals, or indirectly by certain oxidative modifications they cause on key biological molecules (e.g., DNA damage, lipid/protein peroxidation, etc.).

Man-made EMFs do not possess high enough energy to generate free radicals, e.g., on freely moving single H_2_O molecules by a single photon. However, the individual EMF of such photons are fully synchronized (in terms of frequency, polarization, phase, and propagation direction), thereby producing cumulative macroscopic electric and magnetic fields and electromagnetic radiation (EMR). Nonetheless, these may be additively high enough to break covalent bonds and may directly generate free radicals. Secondly, concentrations of naturally occurring free radicals can increase by the prevention of either (i) reactions between them (e.g., appearing as the aforementioned reactants ^•^NO + O_2_^•−^ and products ^•^OH + ^•^NO_2_), or (ii) the reassociation of free radical pairs generated enzymically as transition states. Such prevention can be assisted by the EMF-induced free radical pair mechanism [86,87]. Here, EMFs can prevent the reassociation of free radical pairs by reversing the spin direction of the single electron in one of these free radicals by flipping the direction of its magnetic field component. Thus, the magnetically affected free radical pair ends up consisting of two free radicals, the electron spins of which have become parallel, thereby preventing their re-binding and indirectly increasing their concentration. The free radical pair mechanism has been accounted for by the International Agency for Research on Cancer for the classification of the RF EMFs in the Group 2B category of “possibly carcinogenic to humans” [88]. These two mechanisms of free radical-concentration increase corroborate with the preliminary finding that ELF EMFs increase the concentration of O_2_^•−^ by many fold in various organs of mice exposed to the ICNIRP limit of 100 µT at 50 Hz (pending publication by Dr. Georgiou’s lab).

Therefore, methods for the in vivo specific detection of the key biological free radicals ^•^OH and O_2_^•−^ ([89,90]) are needed in order to unequivocally prove the generation of carcinogenic OS by EMFs.

## Figures and Tables

**Figure 1 ijms-22-10041-f001:**
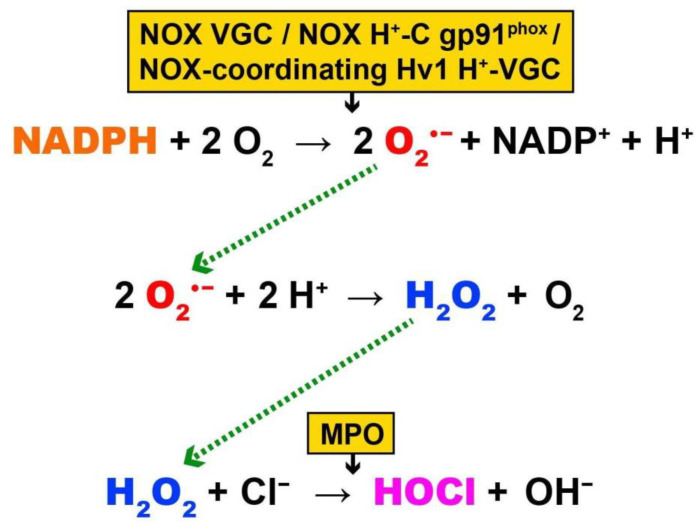
Superoxide radical (O_2_^•^^−^) generated by NOX VGC/NOX H^+^-C gp91^phox^/NOX-coordinating H^+^-VGC Hv1, which is dismutated to H_2_O_2_, which is then used by myeloperoxidase (MPO) as substrate for the two-electron oxidation of Cl^−^ to cytotoxic ROS hypochlorous acid (HOCl).

**Table 1 ijms-22-10041-t001:** Reactions involved in the biochemistry of ROS.

Fe^2+^/Cu^+^ + H_2_O_2_ → Fe^3+^/Cu^2+^ + ^•^OH + HO^−^ (1)
Fe^3+^ + O_2_^•−^ → Fe^2+^ + O_2_ (2)
Fe^3+^ + H_2_O_2_ → Fe^2+^ + H^+^ + HO_2_^•^ (3)
2 Fe^3+^ + AH_2_ → 2 Fe^2+^ + A + 2 H^+^ (4)
AH^−^ + H_2_O_2_ → A^•−^ + H_2_O + ^•^OH (5)
NO + O_2_^•−^ + H^+^ → ONOOH (6)
ONOOH → ^•^OH + ^•^NO_2_ (7)

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
