# Peer review of "Oxidative Stress and NADPH Oxidase: Connecting Electromagnetic Fields, Cation Channels and Biological Effects"

_ijms, 2021, doi:10.3390/ijms221810041_

Round 1
Reviewer 1 Report
The manuscript by Georgiou and Margaritis represents a review article focusing on the connection of oxidative stress with EMF, cation channels and biological effects. It consists of 4 parts focusing on EMF effects on membranes, biochemistry of ROS, biological effects of EMF-induced OS and “EMF vs OS perspectives”. Thus the authors describe some aspects of EMF-induced OS effects but the aims of the whole manuscript are unclear. Moreover, a recent comprehensive review on a similar topic has been published recently by IJMS (10.3390/ijms22073772). Thus I would like to suggest authors thinking seriously on how to focus and present their work suggesting clear and interesting conclusions and further perspectives.
Other comments:
- The manuscript is written in an uncommon manner missing an introduction and conclusions. A lot of work has to be done aiming restructure the content of the manuscript including the abstract to traditional standards making clear the main ideas presented within the manuscript.
- In my opinion, it is better to use ‘Electromagnetic fields’ within the title instead of abbreviation (EMF).
- Typos should be corrected.
Author Response
Dear Reviewer,
Thanks so much for your comments. Please see the detailed revision in the attached profile.
Thanks.

Reviewer 2 Report
The review entitled “Oxidative stress: connecting EMF, cation channels and biological effects” covers the aspects of electromagnetic field (EMF) influence on biological membranes leading to the induction of oxidative stress. Authors describe the following issues such as biochemistry of reactive oxygen species (ROS), EMF-induced oxidative stress, and biological effects and conclude with EMF vs OS research perspectives. Generally, the review is built competently and logically and has a significance for the scientific community. As a reviewer, I have no significant remarks to scientific character of the paper.
In order to improve the manuscript, the following suggestions should be taken into account by the authors:
- Please, delete the references in the abstract of review.
- Abstract: duplication of information is present. Lines 3-4 in abstract and lines 8-9 are identical.
- Abstract: Electromagnetic fields instead of Electrmagnetic fields (letter is missed).
- For better understanding, authors should include the specific sources of Electromagnetic Fields that affect biological membranes.
- Table 1, equation 1: the number of hydrogen atoms in the left and right sides is not equalized.
- The references in the text should not be bold as, for example [31].
After this minor revision, I recommend the present manuscript for publication.
Author Response

(The authors gave the same response as above.)

Reviewer 3 Report
The current article entitled ‘Oxidative stress: connecting electromagnetic fields, cation channel pieces of machinery and biological effects’ focuses on an important topic. But this seems to be a mini review. This paper needs to include some tables and figures that will focus on the research works with their findings, used models etc. I would suggest preparing graphical illustration for each point, for example, EMF effect on membrane function and cation channels can induce OS, OS generation by EMF disturbance of membrane function, Biochemistry of ROS, EMF-induced OS, and biological effects, Theory and research perspectives on conclusive linking of EMFs with ROS/RNS. Please draw cellular machineries and illustrate clearly all the phenomena mentioned in the text. Also prepare a graphical abstract.
Please check lines 306-308.
Author Response
We thank the reviewer for his/her comments, but we believe that they bypass the actual thematic focus of our review, which is clearly presented in its Abstract: “In this review we focus on the main mechanisms of OS generation by EMF-sensitized NADPH oxidase (NOX).” This specifically defined theme shows that our article by no means is a mini review, regarding a wider theme of EMFs, cation channels, and OS association. In actuality, our review uses the most well studied NADPH oxidase cation channel as the best available model for those unknown EMF-exposed cation channels that are directly associated with OS. The reason for this selection is that NADPH oxidase is the only known membrane cation channel, which also possesses as its main function the production of free radicals (the superoxide radical), and by extension it is directly associated with OS as it relates with EMF exposure. In contrast, the reviewer’s proposed thematic suggestions (EMF effect on membrane function and cation channels can induce OS, OS generation by EMF disturbance of membrane function, Biochemistry of ROS, EMF-induced OS, and biological effects, Theory and research perspectives on conclusive linking of EMFs with ROS/RNS), actually recommend for us to write another kind of review article, which addresses all cation channels in general, given that most/near all are not directly related with the production of free radicals and oxidative stress. Besides, the proposed theme for our review article has already been addressed by past review articles.
In reference to the comment “Please check lines 306-308”, unfortunately we were not able to address it because the MS version we downloaded from the journal’s link is not line-number-marked.
Reviewer 4 Report
I find this review interesting, scientifically justified, well written and detailed. The literature is up to date and thoroughly investigated. The review, in general, represents an original, complete and well-planned research. My overall impression is that the authors did a great job in the research of oxidative stress.
Author Response
We thank the reviewer for his very positive comment
Round 2
Reviewer 1 Report
Unfortunately, I cannot conclude that I am satisfied with the authors’ response due to the fact that my concerns have not been addressed properly.
Author Response
We thank, reviewer 1 for reconsidering the revised MS. However, we are unable to respond because his/her comments are not specifically stated. However, we revised the initial MS by addressing his/her main objection, referring to “restructure the content of the manuscript”, adding the missing Abstract section, and including the suggested reference (10.3390/ijms22073772, added as reference #1 in our revised MS).